# Prevalence of reproductive tract infections including sexually transmitted infections among married women in urban and peri-urban mid to low socioeconomic neighbourhoods of Delhi, North India: an observational study protocol

Neeta Dhabhai,[1] Ritu Chaudhary,[1] Teodora Wi,[2] Gitau Mburu,[2]
Ranadip Chowdhury ![ORCID],[1] Deepak More,[1] Leena Chatterjee,[3] Devjani De,[3]
Rita Kabra ![ORCID],[2] James Kiarie,[2] Ndema Habib,[2] Arjun Dang,[3] Manvi Dang,[3]
Sarmila Mazumder ![ORCID] [1]

**Correspondence to**
Dr Sarmila Mazumder;
sarmila.mazumder@sas.org.in

## ABSTRACT

**Introduction** The Global Health Sector Strategy on sexually transmitted infections (STIs), endorsed by the World Health Assembly in 2016 aims to end STIs as public health threat by 2030. WHO conducts global estimates of prevalence to monitor progress towards achieving the same. However, limited laboratory confirmed data exist of STIs and reproductive tract infections (RTIs) apart from few prevalence surveys among key populations and clinic-based reports, including in India. Syndromic approach is the cornerstone of RTI/STI management and to maximise the diagnostic accuracy, there is a need to determine the main aetiologies of vaginal discharge. This study aims to estimate the prevalence of common STIs and RTIs and their aetiological organisms in symptomatic and asymptomatic women living in the urban and peri-urban, mid to low socioeconomic neighbourhoods of Delhi, North India.

**Methods and analysis** A cross-sectional study will be conducted among 440 married women who participated in the 'Women and Infants Integrated Interventions for Growth Study (WINGS)'. Information on sociodemographic profile, sexual and reproductive health will be collected, followed by examination and collection of vaginal swabs for nucleic acid amplification tests to diagnose *Neisseria gonorrhoeae*, *Chlamydia trachomatis* and *Trichomonas vaginalis* and microscopy to identify bacterial vaginosis and *Candida albicans*. Treatment will be as per the syndromic approach recommendations in the Indian National Guidelines. Data will be analysed to estimate prevalence, presence of symptoms and signs associated with laboratory confirmed RTIs/STIs using STATA V.16.0 (StataCorp).

**Ethics and dissemination** This study protocol has been approved by the ethics review committees of the WHO and Society for Applied Studies (SAS/ERC/RHR-RTI/STI/2020). Approval has been obtained by the WINGS investigators from SAS ethics research committee to share the contact details of the participants with the investigators. The findings will be published in peer-reviewed journals and disseminated through scientific conferences.

**Trial registration number** CTRI/2020/03/023954.

## Strengths and limitations of this study

► Recent data on the prevalence of reproductive tract infection (RTI)/sexually transmitted infection (STI) and aetiologies of symptomatic RTI/STI in India are limited; this study responds to this research gap.

► It will provide crucial information on prevalence of laboratory confirmed *Neisseria gonorrhoeae*, *Chlamydia trachomatis* and *Trichomonas vaginalis* among asymptomatic woman, in addition to those with symptoms.

► This study addresses the WHO's recommendation to periodically ascertain the aetiology of vaginal discharge syndrome every 2–3 years in order to update or modify the national guideline.

► This is not a community-based prevalence survey, which would require randomised sampling and would be costly.

► Participants in this study are limited to a prespecified cohort of women hence, generalisability of the findings and conclusions will need to be interpreted with care.

## INTRODUCTION

Sexually transmitted infections (STIs) and reproductive tract infections (RTIs) are a significant cause of global burden of disease.[1] Of the eight pathogens of highest public

health importance, four are curable, *Treponema pallidum, Neisseria gonorrhoeae* (NG), *Chlamydia trachomatis* (CT) and *Trichomonas vaginalis* (TV) while the other four of viral aetiology are not, hepatitis B, herpes simplex virus, HIV and human papillomavirus.[2 3] Globally, 374 million new infections of the four curable STIs occur annually.[2 3] These infections have significant public health consequences including infertility,[4 5] ectopic pregnancies,[6 7] prematurity and neonatal deaths,[8] as well as increased risk of HIV acquisition.[9 10] Among RTIs, bacterial vaginosis (BV) is of emerging importance as it reflects the dysbiotic state of the vaginal microbiome and increases susceptibility to other STIs.[3 4] Candidial genital infection is the leading cause of fungal vulvo-vaginitis and asymptomatic microorganism colonisation can occur in up to 60% of the cases.[8]

Given the importance of STIs and RTIs, strengthening epidemiological data to inform management and control is imperative. In its global health strategy for the control of STIs and tool for strengthening surveillance of STIs, the WHO emphasises on four key elements, including case reporting, prevalence assessment, aetiologies and monitoring of antimicrobial resistance.[11] Global control and prevention of STIs will contribute to progress towards achievement of the sustainable development goals to ensure universal access to sexual and reproductive healthcare services.[2]

To address the limited national data, WHO developed the STI spectrum model to assist countries to estimate the prevalence. Recent estimates suggest that incidence of STIs has plateaued at 1 million cases annually[9]; however, significant regional heterogeneity regarding the burden and availability of data exists especially in Asia and Oceania.[9] In India, there are no recent data to determine the magnitude of STIs. Few prevalence surveys among key populations and pregnant women exist to inform national STI estimates. Regular case reporting of infections such as gonorrhoea is available, but are under-reported, thus the generation of more accurate prevalence data among the general population is needed. Apart from regular case reporting, WHO recommends studies to determine local aetiologies of STIs every 2–3 years to improve management.[10]

In India, it is estimated that 6% of the adult population is infected with one or more STIs at any one time. Due to limited access and affordability of STI laboratory diagnosis at the primary level of care where majority of STI cases seek care, syndromic case management remains the cornerstone of STI control.[12] This, as is the case in many low-income and middle-income countries is partly due to its lower infrastructural/equipment investments and limited availability of expertise. Syndromic approach is based on presence of signs and symptoms reported, and provides immediate treatment for the most common organisms causing the identified syndrome through the use of pre-packaged colour coded drug kits. It is also recommended for the pregnant women.

Based on new evidence, WHO recommends quality-assured molecular assays for treatment of people with symptoms of NG, CT and TV. If quality-assured molecular assays are not available, treatment-based syndromic approach is recommended.[13] However, syndromic approach has certain limitations. It results in over treatment or missed diagnosis of cervical infections (NG and/or CT) compared with aetiological approach using molecular assay.[14 15] While aetiological determination of STI is more accurate, it is often not feasible in most health facilities. Aetiological data among symptomatic and asymptomatic women are inadequate across India.[14] Of the few studies that have explored aetiologies of vaginal discharge, the focus has been among sex workers in 2003 and 2011,[14 15] and these studies found high rates of STIs, especially NG and CT. In 2009, a comparative study of syndromic and aetiologic diagnosis of STIs was conducted among general population in Delhi[16] that included symptomatic and asymptomatic pregnant and non-pregnant women. It revealed a high proportion of women complaining of vaginal discharge, self-reporting of morbidity (65%) and presence of sexually transmitted disease-related syndrome (71.4%). The aetiological microorganism was established in 32.2% of the cases. Our study intends to build on these findings to generate more detailed quality-assured data related to aetiologies of RTI/STIs among symptomatic and asymptomatic women.

## METHODS
### Aims and objectives
The overall aim of the study is to generate information and data related to RTI/STIs among women in India. More specifically, there are three objectives. First, to determine the prevalence of NG, CT, TV, BV and *Candida albicans* (CA) among women residing in the peri-urban and urban mid to low socioeconomic areas of South Delhi at least 14 days after they exit from Women and Infants Integrated Interventions for Growth Study (WINGS). Second, to assess the diagnostic validity of the current RTI/STI syndromic case management practice relative to the aetiologic diagnosis based on laboratory confirmation of NG, CT, TV, BV and CA among symptomatic and asymptomatic women. Third, to identify demographic, behavioural, sexual and clinical factors associated with laboratory confirmed RTI/STI among women. Similar data on syphilis are already being collected as a part of WINGS.[17]

### Study design and participants
This is a cross-sectional independent study enrolling women at least 14 days after their exit from WINGS. WINGS was an individually randomised factorial trial to assess impact of a package of community level health, nutrition, WASH and psychosocial care interventions delivered in preconception, during pregnancy and in the first 2 years after birth, on birth outcomes and infant growth. Thirteen thousand and five hundred married women aged 18–30 years, living with their husband with no or one child, and wanting more children, were enrolled and randomised to receive a pre-conception intervention

package or routine care, until pregnant or 18 months post-enrolment. At pregnancy confirmation, second randomisation was done to receive pregnancy and early childhood interventions or routine care. This resulted in four groups of women that received (1) preconception, pregnancy and early childhood interventions, (2) only pre-conception interventions, (3) only pregnancy and early childhood intervention and (4) no interventions.

Women will be exiting WINGS at two time points: at 18 months since enrolment for women who do not get pregnant, or at 24 months postdelivery for women who get pregnant. A more detailed protocol for WINGS was published elsewhere.[17]

### Sample size
Overall, 6% of adults in India are estimated to have STIs.[14] Existing studies suggest that 32% of cases have laboratory confirmed STI.[16] Therefore, a sample size of 440 women will be adequate to ascertain the prevalence of aetiological diagnosis of STI in symptomatic and asymptomatic women with 95% confidence, 15% relative precision and 20% non-response rate allowance.

### Study procedures
Study preparatory activity started in January 2021 and is expected to be completed in December 2022.

### Recruitment
Study participants who exit from WINGS will be contacted after at least 14 days to ascertain their willingness to participate and those who agree will be offered to take part in the study.

### Informed consent
Written individual informed consent will be obtained from all study participants at enrolment for quantitative data collection, collection of endocervical swabs and for interviews by research assistants at the study clinic. For those who are unable to read, the consent form will be read out by the worker. Those participants who are unable to sign, a thumb imprint will be taken and witnessed (counter signed) by an impartial literate witness. The informed consent form will be translated into simple Hindi language that can be easily read and understood.

### Inclusion criteria
Women who exit WINGS study, give consent and are eligible for sampling. Eligibility is ascertained through screening. Women are screened out and visit rescheduled if they are menstruating at the time of screening, have history of vaginal douching, usage of any vaginal tablets or cream and had sexual intercourse in the last 48 hours. Blood, semen or medication can potentially interfere with test results.

### Exclusion criteria
Women who exit WING study but decline to consent.

### Data collection
Using a structured questionnaire, the research assistants will conduct face-to-face interviews to collect information on sociodemographic, economic and sexual behaviour. The sociodemographic and economic questionnaire includes information on educational status, occupation, family size and structure, annual income, number of family members, religion, household assets, health insurance cover and WASH practices. Questionnaire on sexual behaviour includes information on age of first intercourse, number of partners, use of barrier methods, symptoms of RTI/STI in previous 1 week, history of RTI/STIs and treatment received during last 3 months, among other data.

### Gynaecological examination
A thorough history and genital and pelvic examination including abdominal palpation will be done by the study gynaecologist with aseptic precautions in complete privacy, for all participants irrespective of the symptoms. With the participant in the dorsal lithotomy position, a visual inspection of the external genitalia, followed by a per speculum and bimanual examination will be done.

Signs of RTI/STI and other observations will be documented in an electronic tracker. These will include nature of vaginal discharge (ie, colour, odour, consistency, amount), presence and type of cervical discharge, condition of cervix, any genital ulcers, adnexal or pelvic tenderness, size of uterus, any adnexal mass or lump. High vaginal swab and endo-cervical swab will be collected as per the standard operating procedures. Women will be provided free medication according to the diagnosis, as per the National Guidelines on RTI/STI.[18]

### Sample collection, storage and transportation
Two swabs will be collected, one vaginal and other endo-cervical, by the study gynaecologist. The vaginal walls and cervix will be visualised using self-retaining sterile Cusco's speculum. Vaginal swab specimen will be collected from the posterior fornix and the lateral walls of the vagina for detection of BV and CA and kept in a pre-labelled dry collection tube. Following this, the endocervical swab from the test kit will be inserted gently into the endocervical canal, rotated and withdrawn and kept in the pre-labelled transport reagent container for nucleic acid amplification test (NAAT) for CT, NG and TV.

All samples will be stored at 2°C–8°C in the laboratory and transportation to the testing laboratory will be done daily, in cool boxes maintained at 2°C–8°C.

### Laboratory procedures
NAAT, a real-time PCR having the highest specificity and sensitivity (98%–100%) among the available conventional tests, will be used to detect CT, NG and TV.[19] We will use cartridge based (CB) NAAT Gene Xpert platform which is an automated in vitro diagnostic (IVD) test for qualitative detection and differentiation of DNA of CT, NG and TV and is performed on Cepheid GeneXpert Instrument

Systems.[20] With this we will use the GeneXpert CT/NG Cepheid IVD, and GeneXpert TV Cepheid IVD kits.[21]

The GeneXpert Instrument Systems automate and integrate DNA extraction, amplification and detection of the target sequences in simple or complex samples using reverse transcription (RT-PCR). The systems consist of an instrument connected with a computer, with preloaded software for running tests and viewing and generating results. The systems require single-use disposable cartridges that hold the PCR reagents and host the PCR process.

Reagents for the detection of a Sample Processing Control (SPC), a Sample Adequacy Control (SAC) and a Probe Check Control (PCC) are also included in the cartridge. The SPC is present to control for adequate processing of the target bacteria and to monitor the presence of inhibitors in the PCR.

The SAC reagents detect the presence of a single copy human gene and monitor whether the sample contains human DNA. The PCC verifies reagent rehydration, PCR tube filling in the cartridge probe integrity and dye stability. The primers and probes in the Xpert CT/NG test detect chromosomal sequences in the bacteria. One target is detected for CT (CT1) and two different targets are detected for NG (NG2 and NG4). Both NG targets need to be positive for the Xpert CT/NG test to return a positive NG result. Prepared sample is transferred to the sample chamber of the Gene expert cartridge and loaded on the GeneXpert machine and the test is performed.

Summary and detailed test results are generated by the software in approximately 90 min and are displayed in tabular and graphic formats. Nugent's scoring criteria using a gram-stained slide is the current gold standard with ~77% specificity and 90% sensitivity will be used to diagnose BV.[22]

A smear will be prepared, and gram staining done. Microscopic examination will be done to detect clue cells and semi-quantitative evaluation for gram-positive, gram-negative or gram-variable bacterial morphotypes quantified. Women will be categorised as non-BV (Nugent Score 0–3), intermediate (Nugent Score 4–6) or BV (Nugent Score ≥ 7) (table 1).

Microscopical examination of the vaginal smear will be done for yeast cells with pseudo hyphae to detect CA.

Complete reports will be generated and sent after 48 hours.

### Quality control

For gram stain, on each slide, a control area for *Staph aureus* (*staphylococcus aureus*) (ATCC25923) and *E. coli* (*escherichia coli*) (ATCC25922) is put as a staining control. For CBNAAT, each test kit includes an inbuilt SPC, an SAC and a PCC. External controls (one positive and one negative) will be used in accordance with local, state accrediting organisations, as applicable.

### Kit verification

As the Cepheid kits to be used for the study for CT/NG are Food and Drug Administration (FDA) approved while that

**Table 1** Nugent score

| Organism morphotype | Number/oil immersion field | Score |
|---|---|---|
| Lactobacillus-like (parallel sided, gram-positive rods) | >30 | 0 |
| | 5–30 | 1 |
| | 1–4 | 2 |
| | <1 | 3 |
| | 0 | 4 |
| Mobiluncus-like (curved, gram-negative rods) | >5 | 2 |
| | <1.4 | 1 |
| | 0 | 0 |
| Gardnerella/bacteroides-like (tiny, gram variable coccobacilli and pleomorphic rods with vacuoles) | >30 | 4 |
| | 5–30 | 3 |
| | 1–4 | 2 |
| | <1 | 1 |
| | 0 | 0 |

Total score: 0–3=normal; 4–6=intermediate, repeat test later; 7–10=bacterial vaginosis.

for TV is research use only, a kit verification procedure is planned to evaluate the performance of the GeneXpert platform. A verification panel comprising of 17 samples containing known CT, NG and TV of high, medium and low concentration will be tested on GeneXpert using the recommended kits. The results will be compiled and reviewed to determine the suitability of the platform and the kits for diagnosis of STI in the study.

### Management of RTI/STIs

Women with signs and symptoms of RTI/STIs will be treated based on the syndromic approach as recommended in the national guidelines.[11] Their partners will be notified and provided necessary treatment. The laboratory reports will be reviewed by the study gynaecologists and delivered to their homes with advice to revisit if needed. Asymptomatic women who need further treatment will be invited to the study clinic for an appointment with the study gynaecologists and to receive medicines. Women will also be provided information and counselled on prevention of STI/RTI. Women requiring admission or additional management will be referred to the collaborating hospital. A complete course of treatment and supply of condoms will be provided for free.

### Statistical analysis and outcome measures

Descriptive and inferential statistical analysis will be performed. Simple proportions and 95% confidence bounds will be calculated to determine prevalence for overall laboratory confirmed RTI/STI and individual microorganism by binomial exact method. We will use a method of purposeful selection of covariates to identify variables for the multivariable models.[23 24] In the multivariable models, we will include the variables that will change the OR of the outcome variables by 20% from the univariable models. We

will present the adjusted models, including variables that will be identified in the procedure. We will assess whether any of the demographic, behavioural, sexual and clinical risk factors are modifying the association between laboratory conformed RTI/STI and any of the abovementioned variables on a multiplicative scale. Computation of sensitivity, specificity, predictive values and likelihood ratios will be derived in diagnosing vaginal infection (TV, BV) and cervical infection (NG, CT) using NAAT as the gold standard of diagnosis compared with syndromic approach which is informed by the presence of symptoms and signs. Analysis will be done using STATA V.16.0 (StataCorp).

## Patient and public involvement
Participants and the public were not involved in the design of this study. However, study results will be shared with the participants and informed about their aetiological diagnosis and provided appropriate treatment.

## Confidentiality
All data will be de-identified, analysed and aggregated before reporting. Furthermore, data will be stored securely and only reviewed by the immediate research team. The information collected from the study participants will be kept confidential and stored in a locked area which can only be accessed by the study team. The participants will not be identified by his/her name but only by a study number. All study documents will be stored for a period of 5 years after which they will be destroyed. Knowledge from this study will be disseminated through reports given to the ethics review committee, government agencies and publications but none will have the name of the study participants.

## Tokens/compensation
Participants will be provided with a reasonable token (not exceeding US$2) of appreciation, along with free medical examination, treatment for STI/RTI and condoms. Usually this is given to compensate for the time spent on study participation. It is considered reasonable to compensate participants for their time. The amount is not considered large to constitute an undue inducement.

## Gender and social equity considerations
The proposed study addresses the sexual and reproductive rights of women related to RTI/STI and will identify risks and vulnerabilities in relation to RTI/STIs in the study context. By understanding the prevalence of laboratory confirmed RTI/STIs as well as diagnostic validity of syndromic approach, the study will provide useful information to the national programme. It will also assist women to access treatment, including asymptomatic women, which will be useful in addressing inequality to service access. The research questions are focused on women. However, there will be opportunities for sexual partners of women identified with RTI/STI to be notified and subsequently provided with treatment as per syndromic management following laboratory/aetiologic results.

## Community engagement
This study will enrol women who previously participated in WINGS, which has a large component of community participation. Participants will also be contacted at their homes, and local communities will continually be reached and mobilised with the results to enhance uptake of interventions that promote better woman and infant outcomes.

## Research capacity strengthening
It is envisioned that exchange of information related to conduct of research and data analysis will continually take place between WHO and the research partners.

## Dissemination
The result of the study will be presented in scientific conferences and published in appropriate peer-reviewed journals.

## DISCUSSION
This study aims to fill an existing gap related to a lack of epidemiologic information on STIs in India. A recent systematic review of syndromic approach found that there was very limited validation of syndromic approach as well as aetiologic studies globally.[25] With an ever-changing epidemiology of STIs, there is a need to encourage countries to regularly conduct aetiologic studies of syndromes, to inform national STI case management guidelines.

This study will evaluate the diagnostic validity of RTI/STI syndromic approach among women of reproductive age in India while providing important information related to the prevalence of RTI/STIs and the aetiologies of vaginal discharge syndrome in the study context, where there is insufficient prevalence and aetiologic data. Given the serious consequences of RTI/STIs on women's health, sexual and social relationships and psychological well-being,[26] this research will make an important contribution to the understanding and mitigation of RTI/STIs in India. Data on prevalence will be relevant for improving the national STI case management in India, and for planning RTI/STI services for women, in order to reduce maternal and neonatal morbidity and mortality, adverse birth outcomes and the sequelae of these infections. However the generalisability of the findings from this study to wider population may be limited by the fact that women who participate in research studies self-select, and therefore may bias the prevalence results

In context of high volume of specimens, one of the key challenges for this study will be to ensure quality of sample collection, transport and processing which will be mitigated through training and regular monitoring. While important, our primary intention at this point is not to compare RTI/STI prevalence between two groups of women (those who get pregnant and those who do not). This substudy will not collect data related to syphilis since prevalence data on syphilis is already being collected as a part of WINGS.[16]

**Author affiliations**
¹Centre for Health Research and Development, Society for Applied Studies, New Delhi, India
²Department of Sexual and Reproductive Health and Research, UNDP/UNFPA/ UNICEF/WHO/World Bank Special Programme of Research, Development and Research Training in Human Reproduction (HRP), World Health Organization, Geneve, Switzerland
³Dr Dangs Lab LLP, New Delhi, India

**Acknowledgements** This study is funded by WHO Department of Sexual and Reproductive Health and Research. WHO/SRH department includes the UNDP– UNFPA–UNICEF–WHO–World Bank Special Programme of Research, Development and Research Training in Human Reproduction (HRP). Authors thank Dr Dang's laboratory, implementation partners of the WINGS study (Bills and Melinda Gates Foundation (BMGF)), the department of biotechnology and the Government of India through Biotechnology Industry Research Assistance Council (BIRAC) including CHRDSAS, Safdarjung Hospital and the WHO Department of Maternal, Newborn, Child and Adolescent health (MCA). Views expressed are those of the authors and not necessarily those of their respective institutions.

**Contributors** JK, TW, GM, RK, SM, ND conceived the study. TW, GM, SM, ND, RaC, NH and DM prepared the protocol. All authors reviewed and approved the final manuscript.

**Funding** This study is funded by WHO Department of Sexual and Reproductive Health and Research. WHO/SRH Department includes the UNDP-UNFPA-UNICEF-WHO-World Bank Special Programme of Research, Development and Research Training in Human Reproduction (HRP). Grant number - 202553792. The funding agency has no role in the design of the current protocol and will not have any role in the implementation, analysis and reporting of the study results.

**Competing interests** None declared.

**Patient and public involvement** Patients and/or the public were not involved in the design, or conduct, or reporting, or dissemination plans of this research.

**Patient consent for publication** Not applicable.

**Provenance and peer review** Not commissioned; externally peer reviewed.

**ORCID iDs**
Ranadip Chowdhury http://orcid.org/0000-0002-0055-6653
Rita Kabra http://orcid.org/0000-0001-6595-2035
Sarmila Mazumder http://orcid.org/0000-0003-4200-6112

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
