## [Reviewer comments · BMJ Open]

ARTICLE DETAILS

TITLE (PROVISIONAL)	Prevalence of reproductive tract infections including sexually transmitted infections among married women in urban and peri urban mid to low socioeconomic neighbourhoods of Delhi, North India, an observational study protocol
AUTHORS	Dhabhai, Neeta; Chaudhary, Ritu; Wi, Teodora; Mburu, Gitau; Chowdhury, Ranadip; More, Deepak; Chatterjee, Leena; De, Devjani; Kabra, Rita; Kiarie, James; Habib, Ndema; Dang, Arjun; Dang, Manvi; Mazumder, Sarmila

VERSION 1 – REVIEW

REVIEWER	Fukuda, Yoshiharu Yamaguchi University, Community Health and Medicine
REVIEW RETURNED	24-Dec-2021

GENERAL COMMENTS	The topic is important and the manuscript is well-written. The followings are comments for minor revision. 1. BV in Introduction should be spelled. 2. The inclusion and exclusion criteria could be described more clearly. 3. The questionnaires of socio-demographic, economic and behaviour could be described in more detail.
---

REVIEWER	Gill, Katherine University of Cape Town, IDDM
REVIEW RETURNED	03-Jan-2022

GENERAL COMMENTS	Thank you for the opportunity to review your protocol. STI's have a profound impact on health and are generally under-researched so understanding what is happening on the ground in LMIC is critical. Please address the following questions/ concerns: 1) Please provide more information on the WINGS study. It is not possible to review your protocol without having more information on the parent study and the reader should not have to go searching for this information. 2) Women who have already taken part in a research study may be different to women who have never participated in a study. Please justify strongly why you feel that you can answer a prevalence question from this particular cohort? 3) Please justify why you are only including married women? The title should probably reflect this. 4) Please be careful about drawing bold conclusions from a small convenience sample. This study will provide valuable information but this will only be for a particular group of women in one area at a one-time point. The strengths and limitations section should reflect this.
--

	5) There is no mention of HIV in the protocol. The study team may be missing a valuable opportunity to offer counselling and testing (and treatment) to these women. 6) As the prevalence of drug-resistant gonorrhoea is on the increase worldwide it would have been a good opportunity to understand if AMR is a problem in this area. Please comment on why you have not included this as a study objective? 7) The articles looking at STI prevalence referenced in the introduction are old. If there is nothing newer the authors should highlight this as further justification for their work or include newer literature. 8) Please justify why you are treating syndromically when you will have an etiological diagnosis? Would it not be better to wait for the results and treat with appropriate medication? Syndromic treatment is only justified when you do not have access to lab tests. 9) Will partners be notified for asymptomatic infections?
--	--

VERSION 1 – AUTHOR RESPONSE

Reviewer: 1

Dr. Yoshiharu Fukuda, Yamaguchi University

Comments to the Author:

The topic is important and the manuscript is well-written. The followings are comments for minor revision.

1. BV in Introduction should be spelled.

Thank you for this observation, this was an oversight on our part, and we have corrected it to spell the full form -Bacterial Vaginosis (page 5)

2. The inclusion and exclusion criteria could be described more clearly.

We agree, thank you for the suggestion, we have added the same in the manuscript (pages 9 and 10)

Inclusion criteria – Women who exit the WING study, give consent and are eligible for sampling. Eligibility is ascertained through screening. Women are screened out and visit rescheduled if they are menstruating at the time of screening, have history of vaginal douching, usage of any vaginal tablets or cream and had sexual intercourse in the last 48 hours. Blood, semen and medication can potentially affect the test results.

Exclusion criteria- Women who exit WING study but declined to consent.

3. The questionnaires of socio-demographic, economic and behaviour could be described in more detail.

Thank you for your suggestion, we have now added the detailed description in the manuscript under Data Collection (page 10).

The sociodemographic and economic questionnaires include information on educational status, occupation, family size and structure, annual income, number of members, religion, household assets, health insurance cover and WASH practices. The questionnaire on Sexual behaviour includes information on age of first intercourse, number of partners, use of barrier methods, symptoms of

RTI/STI in previous one week, history of RTI/STIs and treatment received during last three months, among other data.

The full questionnaires have been provided in the supplemental material for Editor (No. 5).

Reviewer: 2

Dr. Katherine Gill, University of Cape Town

Comments to the Author:

Thank you for the opportunity to review your protocol. STI's have a profound impact on health and are generally under-researched so understanding what is happening on the ground in LMIC is critical.

Please address the following questions/concerns:

1) Please provide more information on the WINGS study. It is not possible to review your protocol without having more information on the parent study and the reader should not have to go searching for this information.

Thank you for your suggestion, we have now added a brief description of WINGS on pages 8 and 9 under "Study design and participants"

Women and Infant Integrated Nutrition and Growth Study (WINGS).

WINGS was an individually-randomized factorial trial to assess impact of a package of community level health, nutrition, water, sanitation and hygiene (WASH) and psychosocial care interventions delivered in preconception, during pregnancy and in the first 2 years after birth, on birth outcomes and infant growth. 13,500, married women aged 18-30 years, living with their husband with no or one child, and wanting more children, were enrolled and randomized to receive a pre-conception intervention package or routine care, until pregnant or 18 months post enrolment. At pregnancy confirmation, second randomization was done to receive pregnancy and early childhood interventions or routine care. This resulted in four groups of women who received (1) Preconception, pregnancy and early childhood interventions, (2) only pre-conception interventions, (3) only pregnancy and early childhood intervention, and (4) no interventions.

A protocol describing this study is published elsewhere (Taneja S et al: *Trials* 2020, 21(1):127.) and a study flow diagram is appended in the supplemental material for Editor (No. 4).

2) Women who have already taken part in a research study may be different to women who have never participated in a study. Please justify strongly why you feel that you can answer a prevalence question from this particular cohort?

We thank you for raising this issue. We agree with your concerns that there is a limit to which we can generalise prevalence to women who do not participate in the study. However we note that:

- The study objective in the protocol states that we will examine the prevalence among women who exit from WINGS. The stated objective now is as below

"To determine the prevalence of RTI/STIs including NG, CT, TV, *Candida albicans* and bacterial vaginosis among women, at least 14 days after they exit from WINGS" (under Aims and Objectives on page 8)

- We have highlighted our inability to do a community-based prevalence study (as mentioned in our Strengths and Limitation section – page 4). At the same time WINGS is a large (n=13,500) randomised study with recruitment of women from the community and provides some reassurance that it included a reasonably representative sample of 13,500 women, married and of young reproductive age from the urban low-middle income neighbourhood in South Delhi. The sample of women in this study will therefore provide a fair idea of prevalence among women who participated in

WINGS and to some extent, the study community. Our intention is to take this opportunity of accessibility of the WINGS cohort to attempt to answer important questions about RTI/STIs on prevalence and syndromic validity with minimal expenditure.

• Nevertheless we have acknowledged this limitation by adding the following text in the discussion section on page 17 “Generalisability of the findings from this study to wider population may be limited by the fact that women who participate in research studies self-select, and therefore may bias the prevalence results”

3) Please justify why you are only including married women? The title should probably reflect this.

Thank you for your comment, we have now specified ‘married’ in the title. We have included only married women because this study is being conducted among women who had participated in WINGS. Based on the objectives of WINGS, it included only married women with one or no child, with fertility intention.

4) Please be careful about drawing bold conclusions from a small convenience sample. This study will provide valuable information but this will only be for a particular group of women in one area at a one-time point. The strengths and limitations section should reflect this.

Thank you for your advice, we agree with your observation and have now included the following statement in the section on Strengths and Limitations (page 4)

“As the participants in this study are limited to a prespecified cohort of women in the study community the findings and conclusions need to be interpreted with care”.

5) There is no mention of HIV in the protocol. The study team may be missing a valuable opportunity to offer counselling and testing (and treatment) to these women.

We agree with this viewpoint, but we are unable to remedy it.

In India, as per National Guidelines HIV testing is voluntary, requires pre and post-test counselling in government designated Voluntary Counselling and Testing Centers (VCTCs). As the study is community based and cross sectional in design, it is not equipped to accommodate the HIV testing and counselling.

Moreover, as per the government data the prevalence of HIV in general population of Delhi is low – 0.41%, and 0.13% in ANC clinics,(a surrogate for general population), below the national average i.e. 0.24%. As per this prevalence data and our study sample size (440), we may be able to identify only 1-2 HIV positive participants. (Ref –Delhi State AIDS control Society 2019 – http://colart.delhigovt.nic.in/wps/wcm/connect/doi_dsacs/DSACS/Home/HIV+-+AIDS+Scenario)

6) As the prevalence of drug-resistant gonorrhoea is on the increase worldwide it would have been a good opportunity to understand if AMR is a problem in this area. Please comment on why you have not included this as a study objective?

Yes, we do agree with your statement.

As we understand, ascertaining AMR in gonorrhoea requires bacterial culture and antibiotic sensitivity testing .Setting up bacterial culture and antibiotic sensitivity testing in the study is resource intensive and requires a different laboratory quality assurance system. Collection of gonorrhoea isolates through cultures and transporting the cultures from the community outreach clinic, requires the maintenance of adequate environment for the isolates to be viable.

The collection for cervical specimens for gonococcal cultures requires availability of selective media plates at clinic, inoculated immediately on media, maintenance of CO₂ environment, transportation, to microbiology lab for incubation and further processing within the same day. We are not able to accommodate the complexity and microbiological expertise required at site.

In addition, the yield of positive cultures in asymptomatic women is much lower. India is participating in a Gonorrhoea Antimicrobial resistance Program (GASP) to monitor antimicrobial resistance pattern in *Neisseria gonorrhoeae* among men with urethral discharge, where the yield of positive gonorrhea isolates are higher and is more cost effective, as smaller samples are needed to provide adequate gonorrhea isolates for antimicrobial susceptibility testing.

This study is mainly focused on estimating prevalence of STIs in the study population and compare sensitivity and specificity of Syndromic Approach Vs Lab confirmed diagnosis of STIs.

We hope in future studies we will be able to include it.

7) The articles looking at STI prevalence referenced in the introduction are old. If there is nothing newer the authors should highlight this as further justification for their work or include newer literature.

We agree with you.

After a renewed search, we have not found any appropriate recent prevalence data, and based on your suggestion we have now included the lack of recent prevalence data in the first point under Strengths and Limitations section (page 4).

8) Please justify why you are treating syndromically when you will have an etiological diagnosis? Would it not be better to wait for the results and treat with appropriate medication? Syndromic treatment is only justified when you do not have access to lab tests.

While we agree with your statement that we will have an etiological diagnosis, we are treating women syndromically for the following reasons:

1. It is essential that syndromic treatment is provided on the same day of visit, as this is part of the National STI guideline.
2. Since it will take 48 hours for the results of the etiologic diagnosis to be available (added statement on page 13, under section on laboratory procedures), providing the syndromic treatment will ensure that there are no delays in treatment and decrease risk of onward transmission. This will also avoid missed treatment, as some participants will not return for a visit for the results of the laboratory test. It is however encouraged that all women in the study return for a visit for the laboratory results and additional treatment will be provided based on the etiologic diagnosis.
3. As per the second objective of our study we would like to assess the diagnostic validity of the current RTI/STI syndromic case management relative to the etiologic diagnosis based on the confirmation of NG/CT/TV/BV and CA among symptomatic and asymptomatic women.

9) Will partners be notified for asymptomatic infections?

Yes, for all women who will be treated for infections (asymptomatic and symptomatic) we will follow partner treatment and notification as per the National STI guidelines. We have stated this on page 14, under Management of RTI/STIs.

VERSION 2 – REVIEW

REVIEWER	Gill, Katherine University of Cape Town, IDDM
REVIEW RETURNED	18-Feb-2022
GENERAL COMMENTS	All queries have been satisfactorily resolved.